# Treatment and Outcomes of Metastatic Non-Small-Cell Lung Cancer Harboring Uncommon *EGFR* Mutations: Are They Different from Those with Common *EGFR* Mutations?

**DOI:** 10.3390/biology9100326

**Published:** 2020-10-07

**Authors:** Hyun Ae Jung, Sehhoon Park, Jong-Mu Sun, Se-Hoon Lee, Jin Seok Ahn, Myung-Ju Ahn, Keunchil Park

**Affiliations:** Division of Hematology-Oncology, Department of Medicine, Samsung Medical Center, Sungkyunkwan University School of Medicine, Seoul 06351, Korea; hyunae0608.jung@gmail.com (H.A.J.); sehhoon.park@samsung.com (S.P.); jongmu.sun@samsung.com (J.-M.S.); sehoon.lee@samsung.com (S.-H.L.); jinseok.ahn@samsung.com (J.S.A.); silk.ahn@samsung.com (M.-J.A.)

**Keywords:** uncommon *EGFR* mutation, EGFR-TKIs, T790M, overall survival, progression-free survival

## Abstract

**Simple Summary:**

The present study showed the comprehensive analysis of disease characteristics and treatment patterns in uncommon *EGFR* mutation-positive NSCLC at a major cancer center. This study showed the efficacy of 1G or 2G *EGFR*-TKIs as the 1L treatment, and subsequent therapy including 3G *EGFR*-TKIs in the real-world setting.

**Abstract:**

Approximately 10% of the epidermal growth factor receptor (*EGFR*) mutations in non-small-cell lung cancer (NSCLC) are uncommon *EGFR* mutations. Although the efficacy of second (2G) or third generation (3G) *EGFR* tyrosine kinase inhibitors (EGFR-TKIs) in the patients with uncommon *EGFR* mutation has been proven, further studies are warranted to define the optimal treatment approach for uncommon *EGFR* mutation-positive NSCLC. This study retrospectively investigated the treatment patterns and outcomes of patients with uncommon *EGFR* mutation-positive NSCLC from January 2011 to December 2019 at the Samsung Medical Center, Seoul, Korea. During the study, 2121 patients with *EGFR* mutation-positive NSCLC received first-generation (1G, gefitinib or erlotinib) or 2G EGFR-TKI (afatinib) as the first-line (1L) systemic therapy. Of this, 135 (6.4%) patients harbored uncommon *EGFR* mutations. Of 135, 54 (40%, 54/135) patients had overlapping mutations with major *EGFR* mutations. The objective response rate (ORR) for the 1L EGFR-TKI was 63.3%. The median progression-free survivals (PFSs) were 8.6 months (95% CI: 3.8–13.5), 11.7 months (95% CI: 6.6–16.7), 7.7 months (95% CI: 4.9–17.4), and 5.0 months (95% CI: 3.7–6.1) for major uncommon *EGFR* mutation (G719X, L861Q), compound mutation with major *EGFR* mutation (Del 19 or *EGFR* exon 21 p.L858R), other compound mutation, and other uncommon mutations, respectively. The median overall survivals (OSs) were 25.6 months (16.9–34.2), 28.8 (95% CI: 24.4–33.4), 13.5 months (95% CI: 7.4–27.8), and 9.4 months (95% CI: 3.4–10.5) for major uncommon *EGFR* mutation (G719X), compound mutation with major *EGFR* mutation (Del 19 or *EGFR* exon 21 p.L858R), other compound mutation, and other uncommon mutations, respectively. The response rate, median PFS, and OS were 63.3%, 16.3 months (95% CI: 15.6–16.9), and 37.5 months (95% CI: 35.4–39.6) for common *EGFR* mutation-positive NSCLC. After failing 1L *EGFR*-TKI, repeated tissue or liquid biopsy were carried out on 44.9% (35/78) of patients with T790M detected in 10/35 (28.6%) patients. With subsequent 3G *EGFR*-TKI after failing the first-line *EGFR*-TKI, the ORR and PFS for 3G *EGFR*-TKI were 80% and 8.9 months (95% CI: 8.0–9.8). These patients showed a median OS of 34.6 months (95% CI: 29.8–39.4). The ORR, PFS and OS were poorer in patients with uncommon (especially other compound and other uncommon mutation) than those with common *EGFR* mutations. T790M was detected in 28.6% of the uncommon *EGFR* mutation-positive patients for whom prior 1G/2G *EGFR*-TKIs failed and underwent repeat biopsy at the time of progression.

## 1. Introduction

Exon 19 deletions (Del19) and epidermal growth factor receptor *(EGFR)* exon 21 p.L858R mutations account for approximately 45% and 40% of the cases of epidermal growth factor receptor (*EGFR*) mutation-positive non-small-cell lung cancer (NSCLC). Uncommon *EGFR* mutations account for 7–23% of *EGFR* mutation-positive NSCLCs. Uncommon *EGFR* mutations can be categorized as follows: (i) de novo T790M; (ii) exon 20 insertions; (iii) “major” uncommon mutations Gly719Xaa (G719X), Leu861Gln (L861Q), and Ser768Ile (S768I); (iv) compound mutations; (v) other uncommon mutations [1].

The treatment of *EGFR* mutation-positive NSCLC has been revolutionized with the development of next-generation *EGFR* tyrosine kinase inhibitors (*EGFR*-TKIs). Currently, five *EGFR*-TKIs are available as the first-line (1L) therapy for advanced *EGFR* mutation-positive NSCLC—first-generation (1G) reversible *EGFR*-TKIs, gefitinib, and erlotinib; second-generation (2G) irreversible ErbB family blockers, afatinib, and dacomitinib; third-generation (3G) irreversible *EGFR*-TKIs, osimertinib. Of the *EGFR*-TKI prospective randomized trials undertaken to date, only Iressa Pan-Asia Study (IPASS) [2], NEJ002 [3], and LUX-Lung 2, 3, and 6 [4,5] included patients with uncommon *EGFR* mutations. However, these studies included a small number of cases with uncommon *EGFR* mutations. Thus, it is unclear whether it is the best practice to use 2G or 3G *EGFR*-TKIs as the treatment of choice.

As to which treatment strategy is the best for the patients with major *EGFR*-mutation positive NSCLC, recently, the highly anticipated findings of the phase III FLAURA trial showed 38.6 months of overall survival (OS) with frontline osimertinib, a 3G *EGFR*-TKI, versus 31.8 months with erlotinib or gefitinib [6]. The results of sequential treatment in the GioTag study showed OS and updated time to treatment failure (TTF) analysis of patients with *EGFR* mutation-positive NSCLC who received sequential afatinib/osimertinib. The median OS was 41.3 months (90% confidence interval (CI): 36.8–46.3) in the total population and 45.7 months (90% CI: 45.3–51.5) in patients with Del 19 [7].

While robust data from clinical trials have demonstrated the efficacy, tolerability, and benefits of *EGFR*-TKIs and best sequential treatment, the bulk of these trials were limited to patients whose tumors harbored common *EGFR* mutations.

To the best of our knowledge, there have been no prospective studies on the use of 1G or 2G *EGFR*-TKIs, followed by sequential treatment including 3G *EGFR*-TKI and cytotoxic chemotherapy in patients with uncommon *EGFR* mutation-positive NSCLC. In this study, we describe the real-world data of practice pattern and treatment outcomes in patients with uncommon *EGFR* mutation-positive NSCLC, including objective response rate (ORR), progression-free survival (PFS), and OS.

## 2. Results

### 2.1. Patient Characteristics

From January 2011 to December 2019, 2121 patients with *EGFR* mutation-positive NSCLC received gefitinib, erlotinib, or afatinib as 1L chemotherapy at the Samsung Medical Center. A total of 135 (6.4%) patients harbored uncommon *EGFR* mutations such as G719A/G719C/G719X, L861Q, S781I, H351I, E709K, and de novo T790M. The types of uncommon *EGFR* mutations are described in Table 1. Major uncommon mutations, such as G719X, L861Q, and S791I were 118 (87.4%). Compound mutations were 54 (40%). De novo T790M cases were 15 (11.1%). Other uncommon mutations included L747S and H351I. In this study, there were no patients with *EGFR* insertion 20 who received *EGFR*-TKIs as a 1L treatment. We classified patients with uncommon *EGFR* mutation into the major uncommon *EGFR* mutation (G719X alone (*n* = 61), L861Q alone (*n* = 18), compound mutation with major *EGFR* mutation (Del 19 or *EGFR* exon 21 p.L858R, *n* = 26), other compound mutations (*n* = 13), other uncommon mutations (*n* = 2) and T790M (*n* = 15). There were 47.4% (64/135) male, and ex- or current smokers were 44.4% (60/135). About 25.9% of patients had symptomatic or asymptomatic brain metastasis at initial diagnosis.

### 2.2. Clinical Outcomes of the 1L-Treatment

The ORR (complete response or partial response) for the 1L 1G or 2G *EGFR*-TKIs was 63.3% (76/120). The median PFS was 11.1 months (95% CI: 7.2–15.0) for uncommon *EGFR* mutation-positive NSCLC (Table 2). The median PFSs were 8.6 months (95% CI: 3.8–13.5), 11.7 months (95% CI: 6.6–16.7), 7.7 months (95% CI: 4.9–17.4), and 5.0 months (95% CI: 3.7–6.1) for major uncommon *EGFR* mutation (G719X, L861Q), compound mutation with major *EGFR* mutation (Del 19 or EGFR exon 21 p.L858R), compound mutation (G719X and S791I, G719X and L861Q), and other uncommon mutations (L747S, H835Y), respectively. The median OS were 25.6 months (16.9–34.2), 28.8 (95% CI: 24.4–33.4), 13.5 months (95% CI: 7.4–27.8), and 9.4 months (95% CI: 3.4–10.5) for major uncommon *EGFR* mutation (G719X), compound mutation with major *EGFR* mutation (Del 19 or EGFR exon 21 p.L858R), compound mutation, and other uncommon mutations, respectively. From October 2014 to December 2019, in the uncommon *EGFR* mutation-positive NSCLC group, the median PFS was 15.1 months (95% CI: 12.5–17.7) for afatinib and 7.7 months (95% CI: 1.3–14.1) for gefitinib or erlotinib (Table 2 and Figure 1A) (*p* = 0.165). The median OS was 25.6 months (95% CI: 18.2–33.0) for uncommon *EGFR* mutation-positive NSCLC (Table 2). From October 2014 to December 2019, in the uncommon *EGFR* mutation-positive NSCLC group, the median OS was 34.6 months (95% CI: 16.0–35.2) for afatinib and 15.5 months (95% CI: 9.0–22.0) for gefitinib or erlotinib (Table 2 and Figure 1B) (*p* = 0.03). There was no significant difference in sequential treatment after failing 1L treatment among the three *EGFR*-TKIs.

### 2.3. Clinical Outcomes of the Second-Line Treatment

Among 120 patients (except de novo T790M), 78 experienced disease progression until January 2020. Among them, *EGFR* mutations were evaluated by repeated tissue or liquid biopsy in 44.9% (35/78) patients (Table 3). T790M was detected in 10/35 (28.6%) patients who were successfully biopsied. Among the 78 patients who experienced disease progression, 10 patients received osimertinib as a 3G *EGFR*-TKIs (8/10 patients had overlapping *EGFR* mutations with Del 19 or EGFR exon 21 p.L858R), seven patients who did not have T790M received other *EGFR*-TKIs (e.g., olmutinib), 41 patients received cytotoxic chemotherapy, and 20 patients did not receive any sequential treatment.

In 58 patients who experienced disease progression and received sequential treatment, the median PFS2 was 8.9 months (95% CI: 8.0–9.8) and 4.2 months (95% CI: 2.3–6.2) for third-generation *EGFR*-TKIs and cytotoxic chemotherapy, respectively (Table 3 and Figure 2A) (*p* = 0.05). In 58 patients who experienced disease progression and received sequential treatment, the median OS2 was 15.1 months (95% CI: 9.0–21.2) and 11.0 months (95% CI: 5.1–16.9) for 3G *EGFR*-TKIs and cytotoxic chemotherapy, respectively (Table 3 and Figure 2B) (*p* = 0.214). The median OS was 34.6 months (95% CI: 29.8–39.4), 24.4 months (95% CI: 17.4–31.4), and 5.3 months (95% CI: 2.9–7.7) for 3G *EGFR*-TKIs, cytotoxic chemotherapy, and no sequential treatment, respectively (Figure 2C, *p* < 0.001).

### 2.4. Clinical Outcomes of Patients with De Novo T790M Mutation in the Uncommon EGFR Mutation Group

Among the 15 patients with de novo T790M mutation, 11 patients had an EGFR exon 21 p.L858R mutation and four patients had a Deletion 19 as a coexisting mutation. As a 1L treatment, seven patients received gefitinib, one patient received erlotinib, and seven patients received afatinib. Among them, eight patients received 3G *EGFR*-TKIs after failing the 1L *EGFR*-TKI treatment. The median PFS was 4.9 months (95% CI: 3.8–6.0), and the median OS was 24.0 months (95% CI: 0.7–4.8). Patients who received 3G *EGFR*-TKIs showed a median OS of 38.0 months (95% CI: 10.5–65.4).

### 2.5. Comparison between Common and Uncommon EGFR Mutation

In the uncommon *EGFR* mutation group, there were relatively more male patients and current/ex-smokers than in the common *EGFR* mutation group (Appendix A). During treatment, the re-biopsy rate in the common *EGFR* mutation group was 66.6% (75% between October 2014 and December 2019), and in the uncommon *EGFR* mutation group it was relatively low at 44.9%. Although there was a difference between the two groups in the re-biopsy rate, the T790M detection rate in the patients who underwent re-biopsy was 63.5% in the common *EGFR* mutation group but was 28.6% in the uncommon *EGFR* mutation group. The response rate was 63.3% and 86.6% for the uncommon and common *EGFR* mutation groups, respectively. The median PFS was 16.3 months (95% CI: 15.6–16.9) and 11.1 months (95% CI: 7.2–15.0) for common *EGFR* mutation-positive NSCLC and uncommon *EGFR* mutation-positive NSCLC, respectively (Appendix A) (*p* < 0.001).

In the overall population, the median OS was 37.5 months (95% CI: 35.4–39.6) and 25.6 months (95% CI: 18.2–33.0) for common *EGFR* mutation-positive NSCLC and uncommon *EGFR* mutation-positive NSCLC, respectively (Appendix A) (*p* < 0.001). In the patients who received 3G *EGFR*-TKIs after failing 1L *EGFR*-TKIs, the median OS was 44.4 months (95% CI: 38.9–49.9) and 34.6 months (95% CI: 29.8–39.4) for common EGFR mutation-positive NSCLC and uncommon *EGFR* mutation-positive NSCLC, respectively.

## 3. Discussion

The present study showed the comprehensive analysis of disease characteristics and treatment patterns in uncommon *EGFR* mutation-positive NSCLC at a major cancer center. This study showed the efficacy of 1G or 2G *EGFR*-TKIs as the 1L treatment, and subsequent therapy including 3G *EGFR*-TKIs in the real-world setting.

The median ORR, PFS and OS of patients with uncommon *EGFR* mutation-positive NSCLC were lower than those of patients with common *EGFR* mutation-positive NSCLC (Appendix A). The PFS was not significantly different among the 1G or 2G *EGFR*-TKIs but showed a trend favoring the afatinib group in patients with uncommon *EGFR* mutation-positive NSCLC. There was no significant difference in sequential treatment after failing 1L treatment among the three 1G or 2G *EGFR*-TKIs. However, in comparison to gefitinib or erlotinib, the afatinib group showed statistically superior OS results. Especially in patients with major uncommon *EGFR* mutation (G719X, L861Q), compound mutation with major *EGFR* mutation (Del 19 or *EGFR* exon 21 p.L858R), afatinib showed superior OS results. In our study, less than one-third of the patients with uncommon *EGFR* mutation-positive NSCLC had a T790M mutation in the re-biopsy after failing 1L *EGFR*-TKIs. Most T790M mutation positive cases had overlapping mutations with major (Del 19 or EGFR exon 21 p.L858R) and uncommon *EGFR* mutations. The rate of T790M mutations after failing 1L *EGFR* TKI treatment in our study was only 28.6%, relatively lower than that of common *EGFR* mutation [8,9,10,11]. In a previous study by Yang et al., a lower incidence of acquired T790M mutations was reported in uncommon *EGFR* mutation-positive NSCLCs than in common *EGFR* mutation-positive NSCLCs (27.1 % vs. 45.2%) [12]. It needs to be further prospectively investigated whether the rate of T790M-associated acquired resistance in patients with uncommon *EGFR* mutations after failing EGFR-TKI is lower than in patients with common *EGFR* mutations.

Uncommon *EGFR* mutations are heterogeneous and have various sensitivities to EGFR-TKIs. In addition, the mechanism of acquired resistance to 1L *EGFR*-TKIs has not yet been well defined. Several retrospective studies and case reports of 1G *EGFR*-TKIs showed inconsistent responses in patients with uncommon *EGFR* mutation-positive NSCLC. A post-hoc analysis of prospectively collected data from the participants of the LUX-Lung 2, LUX-Lung 3, and LUX-Lung 6 trials showed the clinical activity of afatinib in patients with advanced uncommon *EGFR* mutation-positive NSCLC, especially G719X, L861Q, and S768I. However, it reported a low activity against T790M and exon 20 insertion mutations. Afatinib showed an ORR of 71% and a median PFS of 10.7 months (95% CI: 5.6–14.7), except for those with T790M or exon 20 insertion mutations, for whom ORR was 9–14%, and PFS was less than 3 months. The median OS was 19.4 months (95% CI: 16.4–26.9) [13]. In a recent afatinib study (*n* = 315), it showed activity against major uncommon mutations (median TTF, 10.8 months; 95% CI: 8.1–16.6; ORR, 60.0%), compound mutations (median TTF, 14.7 months; 95% CI: 6.8–18.5; ORR, 77.1%), other uncommon mutations (median TTF, 4.5 months; 95% CI: 2.9–9.7; ORR, 65.2%), and some exon 20 insertions (median TTF, 4.2 months; 95% CI: 2.8–5.3; ORR, 24.3%) in *EGFR*-TKI naïve patients [1].

In the phase II study of osimertinib for *EGFR*-TKI naïve patients with uncommon *EGFR* mutation-positive NSCLC (KCSG-LU15-09), 22 of the 36 patients received osimertinib as the 1L treatment [14]. A total of 14 patients received osimertinib as a sequential treatment after failing the 1L treatment, except for *EGFR*-TKIs. Osimertinib conferred an ORR of 50%. The median PFS was 8.2 months (95% CI: 5.9–10.5), and the median OS was not reached. Furthermore, the response rate and median duration of response (11.2 months) were lower than those observed with osimertinib in patients with common mutations (Del19, EGFR exon 21 p.L858R). In our study, the median PFS of 1G or 2G generation *EGFR*-TKI was 11.1 months (95% CI: 7.2–15.0), and the median OS was 25.6 months (95% CI: 18.2–33.0) in patients with uncommon *EGFR* mutant-positive NSCLC.

This study has several limitations due to its retrospective nature and single center experience. We had a small number of patients with brain metastasis at initial diagnosis. This study could not evaluate for CNS-specific outcomes or subgroup analysis. Among other things, only 44.9% of the patients were evaluated by repeated tissue or liquid biopsy at the time of progression because there was no accessible site for biopsy (e.g., brain). Furthermore, the polymerase chain reaction-based or direct sequencing methods might have had limitations in detecting compound *EGFR* mutations because of low sensitivity compared to the NGS [15,16]. Success rates for obtaining adequate sample for institutional analysis has ranged from 30% to 80% and is influenced by the quality of the sample, the assay used and resources within institutional molecular laboratories. Initial testing utilized Sanger’s sequencing in isolated DNA from formalin-fixed paraffin embedded (FFPE) tissue but required at least 25% tumor cellularity. Commercial assays such as Therascreen (Qiagen Manchester, U.K.) and Cobas (Roche, Basel, Switzerland), which consider “hot spots” thought predictive of TKI response, are widely utilized. More recently, droplet digital PCR (ddPCR) and next generation sequencing (NGS) coupled with exon-capture strategies have been shown to have increased sensitivity down to 0.01% tumor cellularity. NGS gene panels have improved *EGFR* mutation detection accuracy [17]. It is also well known that the prognosis differs according to the co-existing mutations (*P53*) in common *EGFR* mutation-positive NSCLC. However, in our study, few patients had an NGS result; therefore, further studies are needed to determine the prognostic effect of co-mutations in uncommon *EGFR* mutation-positive NSCLC.

## 4. Materials and Methods

### 4.1. Study Subjects and Data Collection

This study included patients with *EGFR* mutation-positive NSCLC, who started 1L gefitinib, erlotinib, or afatinib treatment for recurrent or metastatic NSCLC at the Samsung Medical Center between January 2011 and December 2019. This non-interventional observational study through big data analysis retrospectively collected de-identified patient data from a clinical data warehouse (CDW) using a unique algorithm with Standard Query Language (SQL) called the ROOT project. Patient demographic characteristics, such as age, sex, smoking history, performance status, and *EGFR* mutation type, were reviewed. We classified the uncommon *EGFR* mutation-positive NSCLC into the group with major uncommon *EGFR* mutations (G719X, L861Q), compound mutations with major *EGFR* mutation (Del 19 or EGFR exon 21 p.L858R), other compound mutations, other uncommon mutations (L747S, H835Y) and T790M. Demographic information was obtained when the 1L *EGFR*-TKI treatment was initiated. *EGFR* mutations were identified using a peptide nucleic acid (PNA)-clamp kit and real-time polymerase chain reaction, COBAS (cobas^®^
*EGFR* Mutation Test v2 [18], or next-generation sequencing (NGS) (sequencing using a cancer panel (CancerSCAN^TM^).

### 4.2. Ethics Statement

This study was reviewed and approved by the Institutional Review Board (IRB) at the Samsung Medical Center (IRB No. 2018-05-130). The trial was conducted following the Declaration of Helsinki (as revised in 2013).

### 4.3. Statistical Analysis

The all-data cut-off date for the analyses was February 2020. PFS and OS were calculated using a Kaplan–Meier estimator and compared using the log-rank test. PFS and OS were presented as median values, with two-sided 95% CIs. PFS was defined as the first date of *EGFR*-TKIs until progression or death resulting from any cause. PFS2 was defined as the first date of sequential treatment after failing 1L *EGFR*-TKIs until progression or death resulting from any cause. OS was defined as the first date of *EGFR*-TKIs until death resulting from any cause. OS2 was defined as the first date of sequential treatment after failing 1L *EGFR*-TKIs until death resulting from any cause. All *p*-values were two-sided, and *p*-value < 0.05 was considered statistically significant. The Korea Food and Drug Administration approved afatinib in October 2014. Among the three 1G or 2G *EGFR*-TKIs, PFS and OS were analyzed in patients who received *EGFR*-TKIs between October 2014 and December 2019.

## 5. Conclusions

The patients with uncommon *EGFR*-mutant NSCLC showed lower ORR, PFS, and OS than the patients with common *EGFR*-mutant NSCLC. In particular, among the uncommon *EGFR* mutation, compared to the group with compounding mutation with major EGFR mutation and the group with major uncommon *EGFR* mutation, the group with other compounding mutation or other uncommon *EGFR* mutation showed inferior outcomes. Compared to common *EGFR* mutation-positive NSCLC, a lower percentage of patients underwent repeated biopsy and showed a lower detection rate of T790M. Less than one-third of the patients received 3G *EGFR*-TKI after failing the 1L *EGFR*-TKIs. However, like the common *EGFR* group, patients with T790M mutation after failing the 1L *EGFR*-TKI who received 3G TKI showed a favorable OS compared to other sequential treatments in the uncommon *EGFR* group. With the availability of more *EGFR* TKIs and a better understanding of tumor biology, further prospective studies are warranted to define the optimal treatment approach for uncommon *EGFR* mutant-positive NSCLC.

## Figures and Tables

**Figure 1 biology-09-00326-f001:**
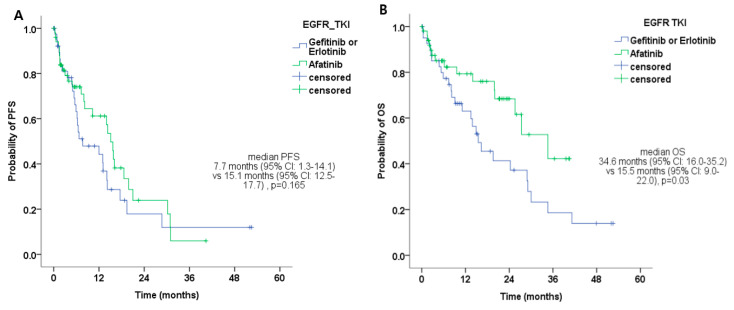
Clinical outcomes of the first-line treatment. (**A**) Median progression-free survival (PFS) according to the type of epidermal growth factor receptor tyrosine kinase inhibitors (*EGFR*-TKIs) in uncommon *EGFR* mutation-positive non-small-cell lung cancer (NSCLC) patients (from October 2014 to December 2019). (**B**) Median overall survival (OS) according to the type of *EGFR*-TKIs in patients with uncommon *EGFR* mutation-positive NSCLC (from October 2014 to December 2019).

**Figure 2 biology-09-00326-f002:**
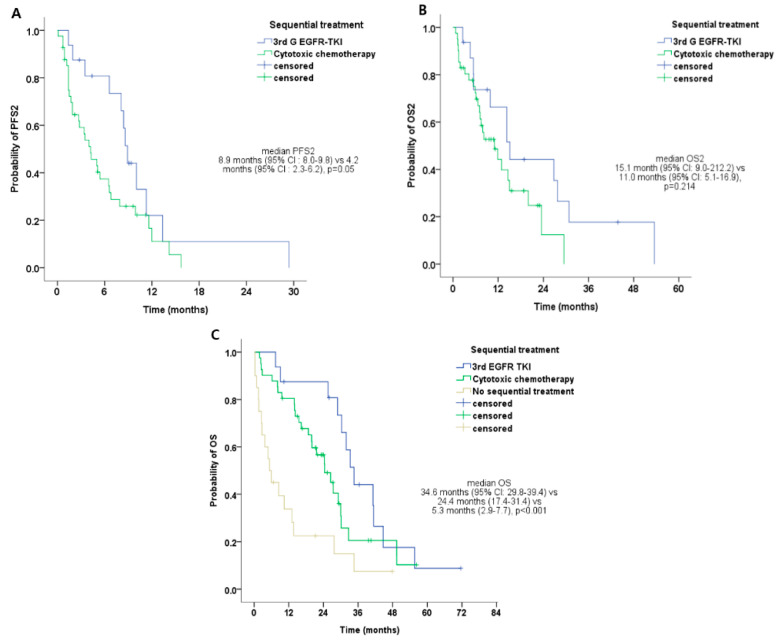
Clinical outcomes of the second-line treatment. (**A**) Median progression-free survival-2 (PFS2) according to sequential treatment in uncommon *EGFR* mutation-positive NSCLC patients. (**B**) Median overall survival-2 (OS2) according to sequential treatment in uncommon *EGFR* mutation-positive NSCLC. (**C**) Median overall survival (OS) according to sequential treatment in patients with uncommon *EGFR* mutation-positive NSCLC.

**Table 1 biology-09-00326-t001:** Baseline characteristics.

Patient Characteristics			Number of Patient (%) Total *n* = 135	
Age	<60 years		47	34.8%
	≥60 years		88	65.2%
Sex	Male		64	47.4%
	Female		71	52.6%
ECOG PS	0		28	20.7%
	1		96	71.1%
	2		11	8.1%
Smoking status	Never smoker		75	55.6%
	Ex-smoker		45	33.3%
	Current smoker		15	11.1%
History of curative thoracic surgery	Yes		26	19.2%
	No		109	80.7%
*EGFR* mutation type	G719X		79	58.5%
		+Del 19	2	
		+L858R	3	
		+S791I	11	
		+E709K	1	
		+L861Q	1	
	L861Q		35	25.9%
		+L858R	16	
		+G719X	1	
	S791I		16	11.9%
		+L858R	5	
		+G719X	11	
	T790M		15	11.1%
		+Del 19	4	
		+L858R	11	
	L747S		1	0.7%
	H835Y		1	0.7%
*EGFR*-TKI as the first-line treatment	Gefitinib		53	39.3%
	Erlotinib		21	15.6%
	Afatinib		61	45.2%

PS, performance status; ECOG, Eastern Cooperative Oncology Group; *EGFR*-TKI, *epidermal growth factor receptor* tyrosine kinase inhibitor.

**Table 2 biology-09-00326-t002:** Clinical outcomes of the first-line treatment.

Uncommon *EGFR* Mutation		PFS		OS	
		11.1 (7.2–15.0)		25.6 (18.2–33.0)	
		Median (95% CI)	*p*-Value	Median (95% CI)	*p*-Value
*EGFR*-TKI *	Gefitinib or Erlotinib	7.7 (1.3–14.1)	0.165	15.5 (9.0–22.0)	0.032
	Afatinib	15.1 (12.5–17.7)		34.6 (16.0–35.2)	

* From October 2014 to December 2019. PFS, progression-free survival; OS, overall survival; *EGFR*-TKI, *EGFR* tyrosine kinase inhibitor.

**Table 3 biology-09-00326-t003:** Clinical outcomes of second-line treatment.

Patient Characteristics		Uncommon Mutation (Except De Novo 790M) (*n* = 120)					
No. of patients who experienced disease progression		78					
Rate of re-biopsy after failing first-line *EGFR*-TKIs		35 (44.9%, 35/78)					
Detection rate of T790M		10 (28.6%, 10/35)					
**Sequential Treatment**	**Number of Patient (%)**	**PFS2**		**OS2**		**OS**	
		**Median PFS2** **(95% CI)**	***p*-value**	**Median OS2** **(95% CI)**	***p*-value**	**Median OS** **(95% CI)**	***p*-value**
3G *EGFR*-TKIs	17 (21.8%)	8.9 (8.0–9.8)	0.055	15.1 (9.0–21.2)	0.104	34.6 (29.8–39.4)	<0.001
Cytotoxic chemotherapy	41 (52.6%)	4.2 (2.3–6.2)		11.0 (5.1–16.9)		24.4 (17.4–31.4)	
No sequential treatment	20 (25.6%)					5.3 (2.9–7.7)	

PFS, progression-free survival; OS, overall survival; *EGFR*-TKI, *epidermal growth factor receptor* tyrosine kinase inhibitor; CI, confidential interval. PFS2 was defined as the first date of sequential treatment after failing 1L EGFR-TKIs until progression or death resulting from any cause. OS was defined as the first date of EGFR-TKIs until death resulting from any cause. OS2 was defined as the first date of sequential treatment after failing 1L *EGFR*-TKIs until death resulting from any cause.

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
