# Peer review of "Treatment and Outcomes of Metastatic Non-Small-Cell Lung Cancer Harboring Uncommon EGFR Mutations: Are They Different from Those with Common EGFR Mutations?"

_biology, 2020, doi:10.3390/biology9100326_

Round 1

Reviewer 1 Report

The manuscript entitled "Treatment and Outcomes of Metastatic Non-Small-Cell Lung Cancer Harboring Uncommon EGFR Mutations: Are They Different from Those with Common EGFR Mutations?" highlighted that the ORR, PFS and OS were poorer in patients with uncommon (especially other compound and other uncommon mutation) than those with common EGFR mutations. 

  • The manuscript may benefit from a grammar and language revision by an English native speaker.
  • Mutations should be reported as follow : gene, exon, p.mutation (e.g. EGFR exon 21 p.L858R)
  • Gene acronyms should be written in italics. Please control.
  • In the discussion section, the Authors should discuss the evidences reported by Gristina et al (PMID: 32113081)
  • In the Methods section the Authors should better describe how mutations were detected. In particular, they should better report the number of patients analyzed by the different techniques, the panel adopted with LOD and reference range.

Author Response

Thanks very much for your comments.

Comments and Suggestions for Authors

The manuscript entitled "Treatment and Outcomes of Metastatic Non-Small-Cell Lung Cancer Harboring Uncommon EGFR Mutations: Are They Different from Those with Common EGFR Mutations?" highlighted that the ORR, PFS and OS were poorer in patients with uncommon (especially other compound and other uncommon mutation) than those with common EGFR mutations. 

  • The manuscript may benefit from a grammar and language revision by an English native speaker.

-> We performed a grammar and language revision by an English native speaker

  • Mutations should be reported as follow : gene, exon, p.mutation (e.g. EGFR exon 21 p.L858R)

-> As your recommendation, we change that.

  • Gene acronyms should be written in italics. Please control.

-> As your recommendation, we change that.

  • In the discussion section, the Authors should discuss the evidences reported by Gristina et al (PMID: 32113081)

-> We added the paper of Gristina et al as a reference and added the comment about the data of Gristinal et al.

  • In the Methods section the Authors should better describe how mutations were detected. In particular, they should better report the number of patients analyzed by the different techniques, the panel adopted with LOD and reference range.

->We added the information of different panel in Method

Reviewer 2 Report

This article by Jung HA treats a topic of high interest both for clinical practice as well as basic research.

The data has been accurately analyzed and it has been presented in a logical and comprehensible structure. The limitations of the work due to its retrospective nature are well acknowledge by the authors during the discussion. I really enjoyed reading the manuscript.

My only suggestion would be to adapt and homogenize the Kaplan-Meier curves from figure 1 and 2. The legend created by SPSS is not neccesary and the font is too small to read. If this figures include the median values, p values and number of patients at risk would be much more informative.

Author Response

Thanks very much for your comments.

Comments and Suggestions for Authors

This article by Jung HA treats a topic of high interest both for clinical practice as well as basic research.

The data has been accurately analyzed and it has been presented in a logical and comprehensible structure. The limitations of the work due to its retrospective nature are well acknowledge by the authors during the discussion. I really enjoyed reading the manuscript.

My only suggestion would be to adapt and homogenize the Kaplan-Meier curves from figure 1 and 2. The legend created by SPSS is not neccesary and the font is too small to read. If these figures include the median values, p values and number of patients at risk would be much more informative.

->Thank you for your constructive feedback. As your recommendation, we revised the figure.

Reviewer 3 Report

Jung et al. have conducted an interesting retrospective study on the treatment and outcome evaluations of metastatic NSCLC harbouring uncommon EGFR mutations. In an effort to further improve the strength of their manuscript, the Authors should take the following action:

Introduction

  • Line 54: report the type and sensibility of the different tests (RT-PCR, NGS, …), and the  available literature data of tissue sample and liquid biopsy evaluations.

Results

  • Line 95: report the number of patients with brain metastasis (symptomatic and asymptomatic) and if undergoing radiation therapy
  • Line 114: clarify why no patients have received osirmetinib therapy
  • Line 136: report all the mutations detected by the repetition of tests and the distribution in the tissue sample/liquid biopsy analysis
  • Line 157: report the data of outcomes in patients with brain metastasis
  • Line185: clarify therapies and outcomes for specific uncommon EGFR mutations
  • Line 195: report the data on patients with brain metastasis in this comparison

Discussion

  • Line 208: clarify for which uncommon EGFR mutations afatinib showed statistically superior OS results

Materials and methods

  • Line 269: report type/platform of the tests (tissue sample and liquid biopsy) and respectively sensibilty

Conclusions

  • Line 291: report and comment the main limitations of this study: retrospective study, sample size, no-standardized  and no-homogenous molecular tests, different sensibility of tests

Author Response

Thanks very much for your comments.

Comments and Suggestions for Authors

Jung et al. have conducted an interesting retrospective study on the treatment and outcome evaluations of metastatic NSCLC harbouring uncommon EGFR mutations. In an effort to further improve the strength of their manuscript, the Authors take the following action:

Introduction

  • Line 54: report the type and sensibility of the different tests (RT-PCR, NGS, …), and the available literature data of tissue sample and liquid biopsy evaluations.

 -> We added the reference of the different test in Discussion session

Results

  • Line 95: report the number of patients with brain metastasis (symptomatic and asymptomatic) and if undergoing radiation therapy

-> Thank you for your constructive feedback. As your recommendation, we added the number of patients with brain metastasis in results

  • Line 114: clarify why no patients have received osirmetinib therapy

-> In Korea, Osimertinib is not approved for the patients with uncommon EGFR mutant NSCLC.

  • Line 136: report all the mutations detected by the repetition of tests and the distribution in the tissue sample/liquid biopsy analysis

-> Thank you for your feedback, we added the detail of mutation test in Discussion

  • Line 157: report the data of outcomes in patients with brain metastasis

-> Thank you for the comment, however, in our study, we had very small number of patients with brain metastasis at initial diagnosis. It is hard to evaluated CNS specific outcomes or subgroup analysis. We added the limitation of our study in the Discussion. We fully agree with your opinion, but we hope you will understand that subgroup analysis is difficult due to the limitations of the current study

  • Line185: clarify therapies and outcomes for specific uncommon EGFR mutations

->Thank you for your constructive feedback. As your recommendation, we added the outcomes for specific uncommon EGFR mutation

  • Line 195: report the data on patients with brain metastasis in this comparison

 -> Thank you for the comment, however, in our study, we had very small number of patients with brain metastasis at initial diagnosis. It is hard to evaluate CNS specific outcomes or subgroup analysis. We added the limitation of our study in the Discussion.

Discussion

  • Line 208: clarify for which uncommon EGFR mutations afatinib showed statistically superior OS results

 -> Thank you for your constructive feedback. As your recommendation, we added the result of afatinib in Discussion session.

Materials and methods

  • Line 269: report type/platform of the tests (tissue sample and liquid biopsy) and respectively sensibility

-> Thank you for your constructive feedback. As your recommendation, we added the information of the EGFR test in Method and discussion.  

Conclusions

  • Line 291: report and comment the main limitations of this study: retrospective study, sample size, no-standardized  and no-homogenous molecular tests, different sensibility of tests

 -> Thank you for your constructive feedback. As your recommendation, we added the limitation of our study. In Discussion.

Reviewer 4 Report

The choice of the best therapy for NSCLC patients carrying uncommon EGFR mutations is a very current topic, especially in light of the increasing use of NGS methods able to providing a high sensitivity and more extensive information about the tumor alterations. This study aims to describe the treatment outcomes (ORR, PFS and OS)  in a large  cohort of patients with uncommon EGFR mutations and known follow-up. As the authors themselves admit, the limit of this work is represented by the small number of cases analyzed with NGS. However, they never
describe how many cases are analyzed by NGS.  A more detailed description of the methods used to detect EGR mutations should be included. This information is important both to understand if the method used to compare cases analyzed with different technologies is
rigorous and if the insertions of exon 20 are absent or have not been searched.

Here are the corrections that I suggest to make in order to improve the description of results and methods.

Line 103: could you indicate if you excluded the T790M cases for the ORR analysis?

From line 105 to line 112: could you insert a table or better summarize the results obtained?

Figure 2: the figure 2C is not present but this is described in the lines 157 and 163.

Line 170: “Among the 15 patients with de novo T790M mutation, 11 patients had an L858R”; could you add that “4 patients had the Del19 as indicated in table1?

Line 189: the supplementary figure is 1A

Line 192: the supplementary figure is 1B

Paragraph 4.1: could you provide a more detailed description of methods to detect EGFR mutations indicating the sensitivity and the DNA region analyzed?

Author Response

Thanks very much for your comments.

Comments and Suggestions for Authors

The choice of the best therapy for NSCLC patients carrying uncommon EGFR mutations is a very current topic, especially in light of the increasing use of NGS methods able to providing a high sensitivity and more extensive information about the tumor alterations. This study aims to describe the treatment outcomes (ORR, PFS and OS)  in a large  cohort of patients with uncommon EGFR mutations and known follow-up. As the authors themselves admit, the limit of this work is represented by the small number of cases analyzed with NGS. However, they never describe how many cases are analyzed by NGS. 

Here are the corrections that I suggest to make in order to improve the description of results and methods.

Line 103: could you indicate if you excluded the T790M cases for the ORR analysis?

->We excluded the T790M cases for the ORR analysis, however, we wrote the outcomes of T790M in different session.

Figure 2: the figure 2C is not present but this is described in the lines 157 and 163.

->We added the figure 2C

Line 170: “Among the 15 patients with de novo T790M mutation, 11 patients had an L858R”; could you add that “4 patients had the Del19 as indicated in table1?

-> Yes, you are right. We added your comment

Line 189: the supplementary figure is 1A

-> Thank you for the comment, we modified the word.

Line 192: the supplementary figure is 1B

-> Thank you for the comment, we modified the word.

Round 2

Reviewer 1 Report

I have no further comments.

Reviewer 3 Report

The manuscript can be accepted with the revision made